# Bisphenol A Modulates Autophagy and Exacerbates Chronic Kidney Damage in Mice

**DOI:** 10.3390/ijms22137189

**Published:** 2021-07-03

**Authors:** Alberto Ruiz Priego, Emilio González Parra, Sebastián Mas, José Luis Morgado-Pascual, Marta Ruiz-Ortega, Sandra Rayego-Mateos

**Affiliations:** 1Division of Nephrology and Hypertension, IIS-Fundación Jiménez Díaz-UAM/IRSIN, 28040 Madrid, Spain; arp.92t@gmail.com (A.R.P.); SMas@fjd.es (S.M.); 2Cellular Biology, Physiology and Immunology Department, Maimonides Biomedical Research Institute of Cordoba (IMIBIC), University of Cordoba, 14004 Cordoba, Spain; jmorgado@uco.es; 3Molecular and Cellular Biology in Renal and Vascular Pathology, IIS-Fundación Jiménez Díaz, Universidad Autónoma Madrid Faculty of Medicine, 28040 Madrid, Spain; mruizo@fjd.es

**Keywords:** Bisphenol A, autophagy, inflammation, oxidative stress, fibrosis

## Abstract

BACKGROUND: Bisphenol A (BPA) is a ubiquitous environmental toxin that accumulates in chronic kidney disease (CKD). Our aim was to explore the effect of chronic exposition of BPA in healthy and injured kidney investigating potential mechanisms involved. METHODS: In C57Bl/6 mice, administration of BPA (120 mg/kg/day, i.p for 5 days/week) was done for 2 and 5 weeks. To study BPA effect on CKD, a model of subtotal nephrectomy (SNX) combined with BPA administration for 5 weeks was employed. In vitro studies were done in human proximal tubular epithelial cells (HK-2 line). RESULTS: Chronic BPA administration to healthy mice induces inflammatory infiltration in the kidney, tubular injury and renal fibrosis (assessed by increased collagen deposition). Moreover, in SNX mice BPA exposure exacerbates renal lesions, including overexpression of the tubular damage biomarker Hepatitis A virus cellular receptor 1 (*Havcr-1*/KIM-1). BPA upregulated several proinflammatory genes and increased the antioxidant response [Nuclear factor erythroid 2-related factor 2 (*Nrf2*), Heme Oxygenase-1 (*Ho-1*) and NAD(P)H dehydrogenase quinone 1 (*Nqo-1*)] both in healthy and SNX mice. The autophagy process was modulated by BPA, through elevated autophagy-related gene 5 (*Atg5),* autophagy-related gene 7 (*Atg7), Microtubule-associated proteins 1A/1B light chain 3B (Map1lc3b/Lc3b)* and *Beclin-1* gene levels and blockaded the autophagosome maturation and flux (p62 levels). This autophagy deregulation was confirmed in vitro. CONCLUSIONS: BPA deregulates autophagy flux and redox protective mechanisms, suggesting a potential mechanism of BPA deleterious effects in the kidney.

## 1. Introduction

Nowadays, there is a great interest of the scientific community to know the mechanisms and effects of endocrine disrupting chemicals in the health of general people and in vulnerable individuals. Among these chemical compounds, 2,2-Bis-(4-hydroxyphenyl) propane (bisphenol A [BPA]) is considered a ubiquitous environmental toxin [1]. This compound has similar structure to the phenolic compounds because of its aromatic rings and can be found as a component of polycarbonate plastics and epoxy resins used to manufacture consumer products, such as dental materials and personal care products [2]. Many clinical and epidemiological studies have described the recurrent exposure of the general population to BPA as well as its harmful effects [3,4,5]. Moreover, the scientific committee of experts on emerging and newly identified health risks (EU-SCENIHR) has recommended to avoid BPA in medical devices [6]. This endocrine disruptor is conjugated by glucuronic acid in the gut wall and liver and excreted by the urine as BPA glucuronide [7]. This exogenous toxin can induce harmful effects in several diseases. In the metabolic syndrome, BPA induced lipotoxicity in non-alcoholic fatty liver disease (NAFLD) [8]. BPA impaired the Glucose transporter type 4 (GLUT4) expression, a key molecule of insulin-induced glucose uptake, aggravating insulin resistance [9], and in obesity BPA increased the body weight and fat levels [10]. In addition, in reproductive disorders BPA affects to testis development inducing toxicity [11]. Importantly, urinary levels of BPA in healthy subjects are associated to the increased risk of coronary heart disease, hypertension and microalbuminuria [12,13]. Chronic kidney disease (CKD) patients are a vulnerable population to BPA exposure [14,15,16,17]. CKD patients present uremic syndrome, characterized by the accumulation of several uremic retention solutes and uremic toxins in the kidney. These uremic toxins can contribute to the progression to end stage renal damage (ESRD) [18,19,20] and to increase the cardiovascular mortality risk. Moreover, in ESRD patients the presence of uremic toxins with big size and bound to proteins induces less effectiveness in the dialysis treatment and renal function impairment [18,21,22]. Some reports described that serum BPA levels are negatively correlated with the annual change in estimated glomerular filtration rate (eGFR), identifying BPA as a renal damage biomarker in diabetic nephropathy and hypertension. In this regard, we have described BPA accumulation in the organism of ESRD patients under chronic hemodialysis with polysulfone dialyzers [15]. Moreover, deleterious effect of high BPA content of polysulfone dialyzers was observed in circulating immune cells of ESRD patients after chronic hemodialysis, characterized by increased oxidative stress and proinflammatory markers [15]. Accordingly, many in vitro studies have described that BPA can induce oxidative stress, mitochondrial dysfunction and inflammation [16,23,24]. However, there is scarce information evaluating whether BPA could affect to experimental CKD progression and the signaling pathways implicated.

Autophagy is a degradative lysosomal process highly conserved in all organisms, that eliminates misfolded and protein aggregates or damaged organelles to maintain intracellular homeostasis and cellular integrity [25]. Importantly, autophagy deregulation has been described in different disease in pathological conditions, such as cancer, metabolic disorders, aging, neurodegeneration and kidney diseases [26,27,28,29,30]. Several studies described the renal toxic effect of BPA, but mechanisms of deleterious effects are not well described.

In this study, we explore the effect of chronic exposition of harmful doses of BPA in healthy or injured kidney and the potential mechanisms implicated in the deleterious effects of BPA in the kidney. To this aim we used an experimental model of chronic kidney damage in mice that resembles the human CKD, the subtotal nephrectomy (SNX) mimicking the progressive renal failure after loss of renal mass in human [31].

## 2. Results

### 2.1. Chronic Systemic Administration of BPA Induces Tubular Damage in the Kidney of Healthy Mice and Exacerbates Kidney Damage in Experimental CKD

To evaluate the effect of BPA administration in mice were dailytrea ted with BPA (120 mg/kg/day intraperitoneal injection (i.p)) and kidney morphology was assessed by hematoxylin-eosin (H&E) and periodic acid-Schiff (PAS) techniques. Healthy mice treated with BPA showed renal lesions, such as inflammatory cell infiltration and tubular dilatation compared to untreated control mice (Figure 1A). In the CKD model of SNX mice presented glomeruloesclerosis and tubular dilatation, as described [32]. The chronic BPA administration for 5 weeks to SNX mice caused an exacerbation of renal damage (Figure 1).

To further investigate BPA effects at molecular level on the kidney, the gene expression of the kidney damage biomarkers *Havcr-1* and *Lipocalin-2*, which codify KIM-1 and NGAL, respectively, were evaluated [33,34]. Both genes were significantly increased in healthy mice treated with BPA compared to untreated mice, starting at 2 weeks of BPA daily exposure in some cases (Figure 1B). Interestingly, *Havcr-1* gene overexpression was higher in BPA than in SNX mice kidneys, remarking the deleterious effect of BPA in tubular cells. Moreover, BPA administration in SNX mice further upregulated the renal expression of both genes, *Havcr-1* and *Lipocalin-2*, compared to SNX untreated mice (Figure 1B). However, the evaluation of renal function (by urea nitrogen (BUN) levels) showed a mild tendency of increase in 5 weeks of BPA compare to control mice (vehicle) (Figure 1C). In contrast, there were no differences between untreated-SNX mice and SNX exposed to BPA group (Figure 1C). These data suggest that BPA exposure to mice induce kidney damage in healthy mice and exacerbated kidney injury in experimental CKD.

### 2.2. Chronic Administration of BPA Causes Inflammatory Cell Infiltration in Healthy and Injured Kidneys by Upregulation of Proinflammatory Factors and Redox-Mediated Processes

The chronic exposure to BPA in healthy mice increased the number of CD3^+^ T lymphocytes infiltrating the renal interstitial space in comparison with untreated control mice (Figure 2A,B), showing an increased tendency along the time, with a marked elevation after 5 weeks of BPA chronic administration. Moreover, BPA treatment in the SNX mice dramatically increased the accumulation of CD3^+^ T lymphocytes in the kidneys compared to untreated-SNX mice (Figure 2A,B), showing an exacerbation of the inflammatory response in injured kidneys exposed to BPA.

The inflammatory cell infiltration in the kidney is regulated by the local production of cytokines and chemokines [35]. In healthy mice exposed to BPA, there was an increased in gene expression levels of several proinflammatory cytokines, such as *Il-6, Ccl-2* (*Mcp-1*) and *Ccl-5* (*Rantes*) compared to control mice, observed after 2 weeks of BPA administration (Figure 2C). Interestingly, *Il-6* and *Ccl-2* gene levels in BPA kidneys were higher than in SNX ones. Moreover, in SNX mice the BPA administration further increased *Ccl-2* and *Il-6* gene expression, showing a key target gene on BPA deleterious effects (Figure 2C). These data support that BPA induces a renal inflammatory response driven by proinflammatory factors upregulation.

Next, whether BPA was able to modulate the nuclear factor (erythroid-derived 2)-like 2 (Nrf2) pathway in the kidney was investigated. Administration of BPA to healthy mice upregulated the renal gene expression levels of *Nrf2*, and its related genes heme oxygenase 1 (*Ho-1*) and NAD(P)H dehydrogenase (quinone 1) (*Nqo1*), showing similar levels to that found in SXN mice kidneys (Figure 3A). Importantly, all those gene levels were synergistically overexpressed in SNX mice exposed to BPA compared to untreated-SNX mice (Figure 3A). Accordingly, protein levels showed a similar tendency in all groups (Figure 3B,C), suggesting an activation of Nrf2 pathway and, consequently, the loss of the protective redox mechanisms in BPA exposed kidney.

### 2.3. Chronic Administration of BPA to Mice Induces Renal Fibrosis in Healthy Mice

The effect of BPA on renal fibrosis was evaluated by Sirius Red staining (Figure 4A,B). This technique disclosed an increase of collagen deposition in a mice exposed to BPA as compared to untreated mice (Figure 4A,B). Moreover, collagen accumulation was markedly elevated in response to BPA exposure in SNX mice compare to SNX group without treatment. Furthermore, the evaluation of protein expression levels of the extracellular matrix protein fibronectin by Western blot showed a significant overexpression of this protein in the kidney of BPA-treated mice compared to healthy mice (Figure 4C).

### 2.4. Stimulation with BPA Induces the Expression of Profibrotic Marker Fibronectin in Human Renal Proximal Tubuloepithelial Cells (HK-2)

To confirm our in vivo results about the fibrotic response induced by BPA, cultured tubular epithelial cells were stimulated with BPA at different doses for 48 h. BPA stimulation induced a dose-dependent increase in fibronectin protein levels (Figure 4D).

### 2.5. Chronic Administration of BPA to Mice Induces the Blockade of Autophagosome Maturation in the Kidney

In order to investigate the effects of BPA administration on autophagy in the kidney, we evaluated the gene expression of key regulators of the autophagy process, such as *Beclin 1*, *Map1lc3b/Lc3b*, *Atg5* and *Atg7*. BPA administration to healthy mice increases gene expression of these autophagy markers, being significant in some cases at 2 weeks and always at 5 weeks (Figure 5A).

In addition, the BPA exposure in SNX mice induced a remarkably overexpression of *Map1lc3b/Lc3b* and *Beclin-1* genes compared to SNX mice. The ratio of LC3II/LC3I protein levels (one of the most important proteins that reflect autophagic flux) was also evaluated by Western blot. BPA administration to mice increased LC3II/LC3I ratio, both in healthy and especially in SNX mice compared to their corresponding controls (Figure 5B). In addition, these results were confirmed by immunofluorescence (Figure 5D).

Another important autophagosome component is p62/SQSTM1, which is recruited by the autophagosome complex acting as a cargo receptor that suffer lysosome degradation during active autophagy process [36]. The p62/SQSTM1levels were increased after BPA exposure in a time-dependent manner in healthy mice (Figure 5C). These results demonstrated that BPA-administration increases autophagosome content by the accumulation of LC3II and but the accumulation of p62/SQSTM1 demonstrated the inhibition of autophagosome-lysosome fusion and selective autophagy substrates accumulation in the kidney.

### 2.6. Stimulation with BPA Induces the Blockade of Maturation of Autophagosome in Human Renal Proximal Tubuloepithelial Cells (HK-2)

To evaluate whether BPA could directly regulate autophagy we performed in vitro studies in human renal proximal tubuloepithelial cells (HK2 cells). Cells were stimulated with different doses of BPA (1, 50 and 100 µM). The autophagy activator Rapamycin was used as positive control. BPA significantly increases LC3II and p62/SQSTM1 protein levels compared with untreated cells (Figure 6A,B), which indicates that autophagic flux is blocked by BPA. To further analyze the effect on the autophagy flux blockade of the effect of 3-methyladenine (3MA), an inhibitor of autophagy and BPA were compared. The accumulation of LC3B and p62/SQSTM1 was higher with BPA compared with 3MA, suggesting that BPA blocks autophagy more efficiently than 3MA.

## 3. Discussion

Nowadays, BPA overexposure generates a controversial discussion about its harmful effect over the population [37]. Different studies have reported that BPA induces reproductive toxicity, abnormal inflammatory disorders and metabolic disease [8,38,39,40]. BPA is considered a ubiquitous environmental toxin in humans, as it is detectable in the urine of most adults and children samples [41,42], because it is mainly removed by the kidneys [43]. Some authors have hypothesized a potential deleterious role of BPA in clinical renal damage suggested by the accumulation of this uremic toxin in patients with CKD when the renal function impair, associated in the vast majority of the cases to increased tubular injury and oxidative stress. This situation is exacerbated in the case of dialysis patients due to the increased BPA levels coming from dialysis membranes [14,44,45]. Preclinical studies have described that BPA is able to induce oxidative stress, mitochondrial membrane depolarization, cell death, lipid peroxidation, as well as abnormal autophagy in experimental lupus nephritis, processes that collaborate to impair renal function, kidney inflammation and damage [46,47,48]. Hence, the main finding of our experimental study is that BPA induces a deleterious effect on the healthy kidney and may contribute to kidney injury progression, through increase oxidative stress, induction inflammatory response, blockage of autophagic flux, and an exacerbated tubular damage, leading to excessive collagen accumulation and kidney fibrosis.

Different studies have shown that the administration of BPA could cause histopathological changes in several tissues, like testis, thymus, liver or heart [49,50,51,52]. In the kidney, a study in a prenatal mouse model predisposed to diabetes mellitus type II (T2DM) showed that the BPA exposure induced glomerular abnormalities and diminished the glomerular formation [53]. Our experiments have demonstrated that BPA administration at a harmful dose of 120 mg/kg/day in healthy C57Bl/6 mice for 2 and 5 weeks can induce renal tissue alterations such as tubular dilatation and inflammatory cell infiltration. These data confirm and extend the deleterious effects of BPA described in previous studies using several doses, exposure times and species. Importantly, in situations or renal impairment, the exposure to BPA results in an exacerbation of the damage, as we showed here in SNX mice, a mouse model that resembles the loss of renal function described in CKD patients, confirming the hypothesis of the harmful effect of this toxin in injured kidneys. Some reports have been described that NGAL and KIM-1 are early prognostic biomarkers of CKD and its progression [54,55]. In 2020, Jacobson et al. [44] developed a prospective clinical study in a cohort of children with CKD to study the effect of BPA and phthalates exposure in the young population, demonstrating that the urinary concentrations of BPA were positively correlated with the increase of NGAL and KIM-1. In this line of study, we focused on analyzing these kidney injury biomarkers at molecular level. As result, we described an exacerbation of the gene expression of KIM1 and NGAL in SNX mice with a chronic exposure to BPA. These observations demonstrated the summative harmful effect of BPA exposure in experimental CKD, and support the use of NGAL and KIM1 as biomarkers of BPA-related kidney injury.

In response to damage, the cells in the organism have the capacity to regulate the ratio between reduction and oxidation (redox balance) of chemical compounds. In a preventive way, the cells use the reactive oxygen species (ROS) (superoxide anions, hydroxyl radicals and peroxides) to modulate gene expression and degrade the ROS excess to finally reduce its harmful effects [23,56]. Higher levels of ROS trigger mutations, exacerbated cell growth, lipid peroxidation, DNA damage, mitochondrial dysfunction and protein modification leading to renal fibrosis and inflammation [57]. Reports in different tissues described that BPA can generate oxidative damage [58,59]. In CKD, the progressive loss of renal function induces clinical dysfunctions and several pathological mechanisms including inflammation, micro-vascular damage, ROS production and fibrosis [60,61,62]. The ROS production in the kidney contributes to cardiovascular events in CKD patients [63]. At the same time, the analysis of human samples, experimental models and cultured cells in response to BPA demonstrate the linkage between inflammation and cardiovascular disease [64,65,66]. On the other hand, experimental hypertension induced by BPA was related with oxidative stress production [67]. In young CKD patients exposed to BPA also has been described the presence of antioxidant response [44]. This renal response against oxidative damage was also observed in renal cells of different species stimulated with BPA [46,68,69,70]. Accordingly, some experimental studies described the pro-oxidant characteristics of chronic exposure to BPA. A study in rats exposed to BPA (oral administration) for 5 weeks showed an increase in nitric oxide and malondialdehyde (MDA) as well as reduced glutathione and superoxide dismutase levels [71]. Similar results in ROS induction were observed in mice orally administered with 120 and 240 mg/kg/day of BPA, at doses similar to our study [48]. In a model of Lupus nephritis in MRL/lpr mice oral administration of BPA decreased Nrf2 and increased NFk-B gene and protein levels and impaired lupus nephritis [47]. Our group previously described that BPA can induce oxidative damage in peripheral blood mononuclear cells (PBMCs) from patients on hemodialysis with polysulfone dialyzer through the increase of Nrf2 transcription factor, and genes that encoding antioxidant proteins such as *Ho-1* and *peroxiredoxin 1* (*Prx-1*) [15]. Accordingly, we found that the administration of BPA to mice produced the strong induction of antioxidant defenses at gene levels such as *Nrf-2*, *Nqo-1* and *Ho-1*, especially in SNX mice, and probably as a consequence of increased ROS production, pointing out the Nrf2 pathway as an important signaling mechanisms modulated by BPA in renal damage.

Some evidences have described that BPA can exert proinflammatory actions. BPA presence in urine of postmenopausal and pregnant women has been associated to the number of white blood cell count (inflammatory cells) and C reactive protein levels [64,72,73]. In vitro studies suggested that BPA increases the production of pro-inflammatory cytokines, such as TNF-α and IL-6 [16,74]. Recently one study developed in rats showed that the oral administration of BPA for 30 days induced an increase in the release of proinflammatory cytokines such as IL-1β, TNF-α and IL-6 as well as the deterioration of renal function [75]. Our results confirm that systemic administration of BPA in mice induces the renal gene expression of proinflammatory and chemotactic mediators such as *Il-6, Ccl-2/MCP-1* and *Ccl-5*/Rantes, and this inflammatory response is aggravated in CKD mice, contributing to poor renal prognosis. In addition, chronic BPA in mice generated an exacerbated recruitment of CD3^+^ T lymphocytes to the renal interstitial space, more marked in injured kidneys, a situation that could contribute to the production and release of more inflammatory mediators, in particular *Ccl-2/MCP-1*.

Macroautophagy/autophagy is an intracellular degradation system that develops a key role in inflammatory response, and the modulation of inflammation through autophagy can be assessed as a potential treatment for damaged kidneys [76]. Several studies demonstrated the main role of autophagy in renal diseases [76,77,78]. LC3-II level reflects the balance between the autophagosome generation and degradation in a dynamic pathway, but cannot determine autophagy flux [79]. Hence, quantification of other proteins to determine autophagy flux is necessary. The p62/SQSTM1 is an autophagic substrate, and another important protein in autophagic flux, not only because it plays an important role in the degradation of the accumulated redundant proteins [80], but also it sequesters aggregated proteins and promotes their degradation [81,82]. Thus, activation of autophagy produces a decrease in protein levels of p62/SQSTM1, and in the opposite way inhibition of autophagy leads to increase of p62/SQSTM1 levels. All of these, increased p62/SQSTM1 indicates disrupted autophagic flux. Actually several reports determined the important role of autophagy in CKD [83,84] and clinical studies described the altered autophagy process in renal patients through the LC3II/LC3I ratio or γLC3. This pathological autophagy modulation was not reversed by hemodialysis treatment [85]. The autophagy process is necessary to maintain the body homeostasis and protein recycling and also is critical in the renal inflammatory response and is regulated by oxidative stress [86]. There is controversial data about the beneficial and deleterious role of autophagic process in renal damage. Studies in renal diseases such as diabetic nephropathy [87], Lupus nephritis [88,89], AKI induced by nephrotoxic [90], glomerular diseases or autosomal dominant polycystic kidney disease [91,92] described the harmful effects of downregulated autophagy [86,93]. In contrast, in a model of CKD such as unilateral ureteral obstruction (UUO), the autophagy overexpression impaired pathological features of renal damage [93]. Accordingly, the autophagy inhibitor 3MA induces cell apoptosis in the tubuli and fibrotic response in the obstructed kidney in a rat model [94]. On the contrary, the increased levels of LC3-II and Beclin-1 during the progression of renal damage induced tubular cell death and atrophy [94,95,96]. Currently, few groups have studied the relationship between autophagy and exposition of BPA in organs, like reproductive system or liver, showing controversial data [47,97,98]. Only one manuscript have evaluated the role of BPA in autophagic flux in experimental Lupus nephritis showing an increase of associated autophagy proteins, such as LC3II [47]. Now, in healthy mice and in a model of progressive CKD, we have found that BPA overexpressed several autophagy genes, increased LC3II/LC3I ratio and p62 renal levels, suggesting that BPA increased autophagosome formation but disrupted autophagic flux by accumulation of p62/SQSTM1. Moreover, our in vitro studies using BPA gradient concentrations as well as pharmacological autophagy modulators, demonstrated the direct effect of BPA on autophagy. In tubular epithelial cells BPA mimic the responses elicited by 3-MA, an autophagy inhibitor, including the up-regulation of the levels of LC3-II and p62, showing that BPA suppresses autophagic flux. In this sense, different studies have observed that inhibition of autophagic flux in the kidney epithelium is enough to trigger a degenerative disease of the kidney [99]. Therefore, exposure to BPA may contribute to degenerative kidney progression as a consequence of blockage of autophagic flux.

Chronic inflammation contributed to the development of tubulointerstitial fibrosis in CKD [100,101]. Several studies described the capacity of BPA to induce fibrosis in different tissues such as endometrium, heart and liver [102,103,104]. In contrast, there were no studies about the effect of BPA on renal fibrosis. Our study showed that in healthy mice the continuous administration of BPA induced an increase of collagen deposition (Sirius red staining) as well as an increase of extracellular matrix proteins such as fibronectin. Similar results were observed here, in the in vitro experiment in cultured renal cells stimulated with several doses of BPA. These data support the idea of BPA as a deleterious compound by the regulation of profibrotic processes leading to renal damage progression, in healthy and injured kidneys as well as in cultured cells.

The data presented in this work exhibit that BPA promote renal inflammatory cell infiltration, by upregulation of cytokines and chemokines, increased oxidative stress and blockage of autophagic flux. To sum up, our work supports the idea that BPA could be an important toxic for patients with CKD and contribute to kidney injury progression.

## 4. Materials and Methods

### 4.1. Reagents

We used the following inhibitors from Sigma Aldrich (San Luis, MO, USA): 3-Methyladenine (3-MA, M9281), Rapamycin (37094), and Bisphenol A (BPA) (239658). They were solubilized in dimethyl sulfoxide (DMSO) and in corn oil (C8267) for in vitro and in vivo experiments respectively.

### 4.2. Cell Cultured Studies

Human renal proximal tubular epithelial cells (HK-2 cell line, ATCC CRL-2190) were cultured in RPMI 1640 (Sigma-Aldrich), supplemented with 10% heat-inactivated fetal bovine serum (FBS), 100 U/mL penicillin, 100 µg/mL streptomycin, 2 mM glutamine, insulin transferrin selenite (5 µg/mL) and hydrocortisone (36 ng/mL, Sigma Aldrich). We then cultured the cells at 37 °C in 5% CO_2_ atmosphere.

A steroid-free medium containing DMSO (0.5% *v/v*) was used as the control. All experiments were performed in complete medium because starvation conditions induce basal autophagy activation. The cells were exposed to the following treatments: 3-Methyladenine (5 mM), Rapamycin, (1 µM; Sigma Aldrich; San Luis, MO, USA). The dose of BPA was established based on previous experience in the laboratory [16].

### 4.3. In Vivo Studies

All procedures on animals were performed according to the recommendations issued by the European Community, including the requirements of Directive 2010/63/EU of the European Parliament and of the Council of 22 September 2010 on the protection of animals used for scientific purposes, including the 3Rs principle. These protocols were approved by Instituto de Investigación Sanitaria Fundación Jiménez Díaz (IIS-FJD) Animal Research Ethical Committee and by the Comunidad de Madrid (PROEX 080/18). Studies were done in adult mice C57Bl/6 mice (12 to 14 weeks, weight 20 g, seven to eight animals per group).

Prior to the surgery, the animals were shaved in the abdominal region to facilitate the surgery procedure and to avoid the hair contamination that could induce infection of mice. The anesthetic that was used to perform surgeries was Isofluorane using vaporizer equipment at optimal dose 4.5% for induction; 1–2% for maintenance/inhalation route and analgesics for 3 days post surgery (buprenorphine 0.1 mg/kg/day by subcutaneous injection) will be used to improve post-operative recovery. At the time of euthanasia, animals were anesthetized with with 5 mg/kg xylazine (Rompun, Bayer AG; Leverkusen, North Rhine-Westphalia, Germany) and 35 mg/kg ketamine (Ketolar, Pfizer; New York, NY, USA) and kidneys perfused in situ with cold saline before removal. Then, kidney portions were fixed in buffered formalin for immunohistochemistry studies or immediately frozen in liquid nitrogen for gene and protein studies. Blood samples were collected to analyze urea plasma levels.

Subtotal nephrectomy (SNX): The model of 5/6 Subtotal nephrectomy (named SNX) was performed through a surgical procedure in two different phases under the effect of inhalatory anesthesia (Isofluorane using vaporizer equipment at optimum dose 4.5% for induction; 1–2% for maintenance) [105]. First, surgery started with an abdominal incision in the right flank of the animal, performed to have access to the peritoneal space, then we localized the right kidney and the renal artery was clamped with a micro-vascular clamp (1–2.25 mm; ref: 70007; S & T; Neuhausen; Switzerland) to cut the blood flow before the ablation the two poles of this kidney. The ablation of the kidney poles was developed by a cut with a scissors and a piece of absorbable gelatin hemostatic sponge (gelita medical; ref: GS950; Eberbach, Germany) was placed to avoid the loss of blood after the reestablishment of the blood flow. This surgery needed 7 days to allow the animal to recover from it. Then, a second surgery was done, characterized by the left nephrectomy through left flank incisions. The process involved passing a surgical silk around the renal artery and making a double knot over it, in order to cut off the flow of blood. After this, the kidney was removed by a cut with scissors. In the sham-operated group mice, a right flank incision was performed at the first stage, both poles of kidney were identified, and then the flank incision was closed. A subsequent left flank incision was performed two weeks later, the left renal artery was identified, and then the flank incision was closed.

Chronic exposure to Bisphenol A. Mice received 120 mg/kg/day body weight BPA dissolved in corn oil by intraperitoneal (i.p) daily injection (from Monday to Friday) and mice were sacrificed 2 or 5 weeks later. Five groups were studied (N = 4–7 per group) classified in two different biological situations:(A).Harmful effect of BPA in healthy mice: (i) vehicle; (ii) BPA at 2 weeks; (iii) BPA at 5 weeks;(B).Harmful effect of BPA in established CKD: (iv) SNX vehicle; (v) SNX with BPA at 5 weeks.

Vehicle mice (group i) and SNX vehicle mice were i.p daily injected (from Monday to Friday) with corn oil for 5 weeks (previous studies of the group demonstrated that corn oil has no harmful effect). In addition, SNX with BPA mice (group v) were subjected to BPA for 5 weeks after the subtotal nephrectomy surgery. The BPA dose was selected to observe a deleterious effect in healthy animals with normal renal function being able to excrete the excess of BPA by the urine. The dose of 120 mg/kg/day was previously described as the median lethal dose (LD50) value [106], to induce ROS production in C57Bl/6 mice in the kidney, as well as damage in several tissues [107,108,109,110]. Furthermore, this dose was defined with lowest observed adverse effect level (LOAEL index) by the European Commission [111].

### 4.4. Gene Expression Studies

For mRNA isolation, we pulverized frozen kidney pieces in a metallic chamber and dissolved them in 500 ul Trizol (Invitrogen, San Diego, CA, USA). We also isolated total RNA from cultured cells with Trizol. The quantity and purity of extracted RNA were assessed by measuring absorbance at 260 nm and 280 nm in a UV spectrophotometer (NanoDrop Inc., Wilmington, DE, USA). Only samples with an Abs260:Abs280 ratio up to 1.8 were considered valid for real-time PCR. We synthesized cDNA utilizing the High Capacity cDNA Archive kit (Applied Biosystems, Foster City, CA, USA) using 2 ng total RNA primed with random hexamer primers following the manufacturer’s instructions. Multiplex real-time PCR was performed using fluorogenic TaqMan MGBprobes and primers designed by Assay-on Demand gene expression products (Applied Biosystems; Waltham, MA, USA): mouse *Havcr1/kim1* (Mm00506686_m1); mouse *Lipocalin2/N-Gal* (Mm01324470_m1); mouse *Nrf-2* (Mm00477784_m1), mouse *Ho-1* (Mm00516005_m1), mouse *Nqo-1* (Mm00500821_m1), mouse *Ccl-2/Mcp-1* (Mm00441242_m1), mouse *Ccl-5/Rantes* (Mm01302428_m1), mouse *Il-6* (Mm00446190_m1), mouse *Beclin1* (Mm01265461_m1), mouse *Map1lc3b/Lc3b* (Mm00458724_m1); mouse *Atg7* (Mm00512209_m1), mouse *Atg5* (Mm01187303_m1). Data were normalized with murine glyceraldehyde 3-phosphate dehydrogenase (GAPDH, assay IDs: Mm99999915_g1 or) and we calculated the mRNA copy numbers for each simple by the instrument software using cycle threshold (Ct) value (“arithmetic fit point analysis for the lightcycler”). We expressed results in copy numbers, calculated relative to unstimulated cells or control mice, after normalization against GAPDH, as described previously.

### 4.5. Protein Studies

Western blotting was used to assess protein levels. Protein extracts from cultured cells were obtained by homogenization and centrifugation as described previously [112]. Then, samples were separated by SDS-PAGE, transferred to polyvinylidene difluoride membranes (Millipore, Bedford, MA, USA), blocked in PBS containing 0.1% Tween- 20, 7.5% dry skimmed milk for 1 h at room temperature, and incubated in the same buffer with different primary antibodies overnight at 4 °C. We used the following antibodies: Nrf2 (1:500, sc-722, Santa Cruz Biotechnology), HO-1 (1:1000, SPA-896; Enzo life sciences; Farmingdale, NY, USA), NQO-1 (1:1000, Sc-32793, Santa Cruz Biotechnology; Dallas, TX, USA), NRF2 (1:500, Sc-365949, Santa Cruz Biotechnology), HO-1 (1:500, Sc-136960, Santa Cruz Biotechnology; Dallas, TX, USA), LC3B (1:1000, NB100-2220, Nobus Biological; Englewood, CO, USA), p62/SQSTM1 (1:1000, 5114S, Cell signaling; Danvers, MA, USA), Fibronectin (1:5000, A2033, Millipore), Tubulin (1:5000, T5168, Sigma; San Luis, MO, USA), GAPDH (1:5000, CB1001, Millipore; Burlington, MA, USA). After washing, we incubated the membranes with peroxidase-conjugated secondary antibody and developed them using an ECL chemiluminiscence kit (Amersham Pharmacia Biotech, Piscataway, NJ, USA). We quantified proteins in all samples using the BCA method, and we loaded 30 mg protein in each lane. The quality of proteins and efficacy of protein transfer were evaluated by Red Ponceau staining). To evaluate equal loading, we stained the membranes with anti-tubulin antibody (Sigma, San Luis, MO, USA). We scanned the autoradiographs using the GS- 800 Calibrated Densitometer (Quantity One; Bio-Rad, Madrid, Spain).

### 4.6. Renal Histology and Immunohistochemistry

Paraffin-embedded kidney sections (3 μm) were stained using conventional methods. Antigen retrieval was performed by PTlink link system (Dako Diagnostics; Agilent technologies, Santa Clara, CA, USA) with sodium citrate buffer (10 mmol/L) adjusted to pH 6–9 depending on the immunohistochemical marker, followed by immunohistochemical staining. Steps: (1) endogenous peroxidase blockade; (2) primary antibodies incubation; anti-CD3; (3) washing; (4) DUOFLEX Doublestain EnVision™ treatment, using 3,3’-diaminobenzidine as chromogen. Sections were counterstained with Carazzi’s hematoxylin. The intensity of the reactive mark was obtained using Image-Pro Plus software. For each sample (processed by duplicate in a blinded manner), the average value was obtained from the analysis of 4 fields (20× objective) as density/mm^2^ or percentage stained area vs. total analyzed area. Data are expressed as fold increase over control mice, as mean ± SEM of 4–7 animals per group. Negative controls include non-specific immunoglobulin and no primary antibody.

### 4.7. Immunofluorescence Staining of LC3B

Paraffin-embedded kidney sections (3 μm) were subjected to Antigen retrieval by PTlink link system. After the slides were blocked with 10% BSA and 10% FBS for 1 h, they were incubated with LC3B primary antibody (1/200; Nobus Biological; Englewood, CO, USA) for 1 h, followed by an AlexaFluor^®^ 488 conjugated rabbit anti-mouse-conjugated secondary antibody (1/200; Invitrogen), for 1 h. Absence of primary antibody was used as negative control. Samples were mounted in prolong gold (Thermo Fisher Scientific; Waltham, MA, USA) and examined using a Leica DM-IRB confocal microscope.

### 4.8. Statistical Analysis

Statistical significance was assessed in R v 4.05 (https://cran.r-project.org/ accessed on 30 June 2021) and graphs plotted using GraphPad Prism 5. Comparison between SNX and SNX + BPA groups was performed using a parametric unpaired two-sided T test based on the normality of the data distribution (tested with a Shapiro–Wilk normality test) and homogeneity of variances (tested with F test). In contrast, the comparisons of WT vs. BPA 2 w and 5 w groups and the in vitro studies were evaluated by one-way ANOVA, followed by the Tukey HSD multiple comparison test based on the normality of the data distribution. The data were presented as mean ± SEM or median of the data and interquartile range (IQR) (from lower (25%) to upper (75%) quartile). Only *p* values ≤ 0.05 were considered significant.

## Figures and Tables

**Figure 1 ijms-22-07189-f001:**
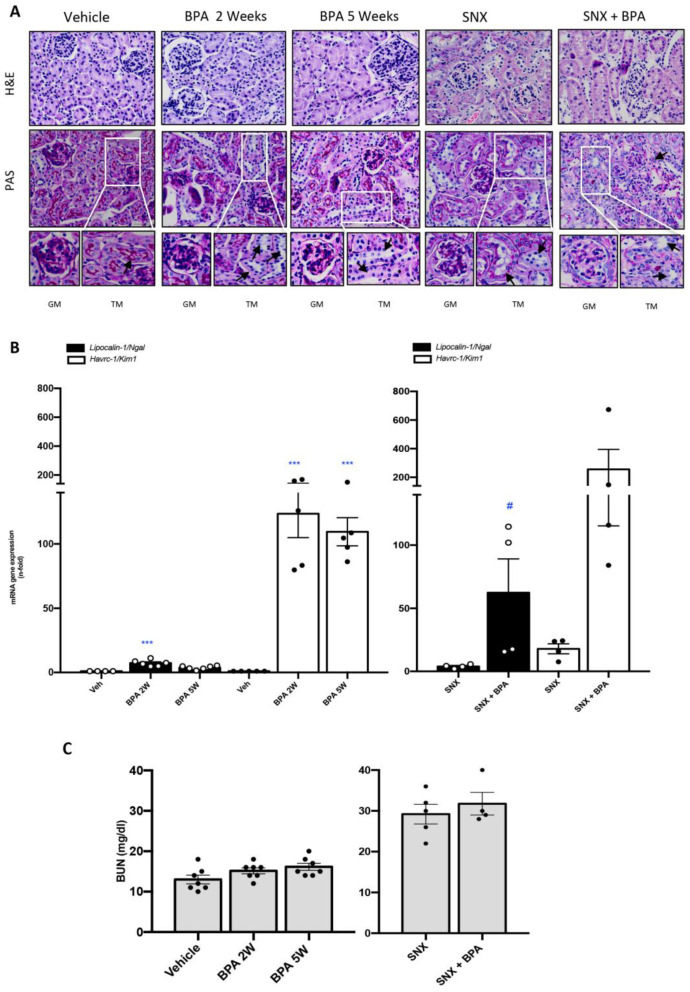
Systemic chronic administration of BPA induces kidney damage in healthy mice and exacerbates renal lesions in an experimental model of progressive CKD. Animals were intraperitoneal (i.p) injected with 120 mg/kg/day BPA (5 days a week) or vehicle (corn oil) and sacrificed after 2 or 5 weeks. Some animals were subjected to subtotal nephrectomy (SNX), then treated or not with BPA and sacrificed after 5 weeks. (**A**) Representative pictures of H&E and PAS staining show tubular dilatation, mesangial matrix proliferation, glomerulus disorganization and loss of glomerular/tubular basement membrane and the brush border of the proximal tubules in animals exposed to BPA mainly in SNX compared to vehicle mice. Magnification 200× (GM: glomerular magnification; TM: Tubular magnification; arrows indicate tubular damage /scale bar appears in lower right area of the image and represents 50 µm). (**B**) Renal mRNA expression of *Havcr-1/Kim-1* and *Lipocalin2* (that codify KIM-1 and N-GAL, respectively) were evaluated by RT-PCR. (**C**) Serum Urea Nitrogen (BUN) levels are shown in mg/dl. All data are presented as the mean ± SEM of 4–7 mice per group; *** *p* < 0.001 vs. Vehicle, # *p* < 0.05 vs. SNX. Comparison between SNX and SNX + BPA groups was performed using a parametric unpaired two-sided *t* test. The analysis of WT vs. BPA-treated groups was done by one-way ANOVA, followed by the Tukey HSD multiple comparison Test.

**Figure 2 ijms-22-07189-f002:**
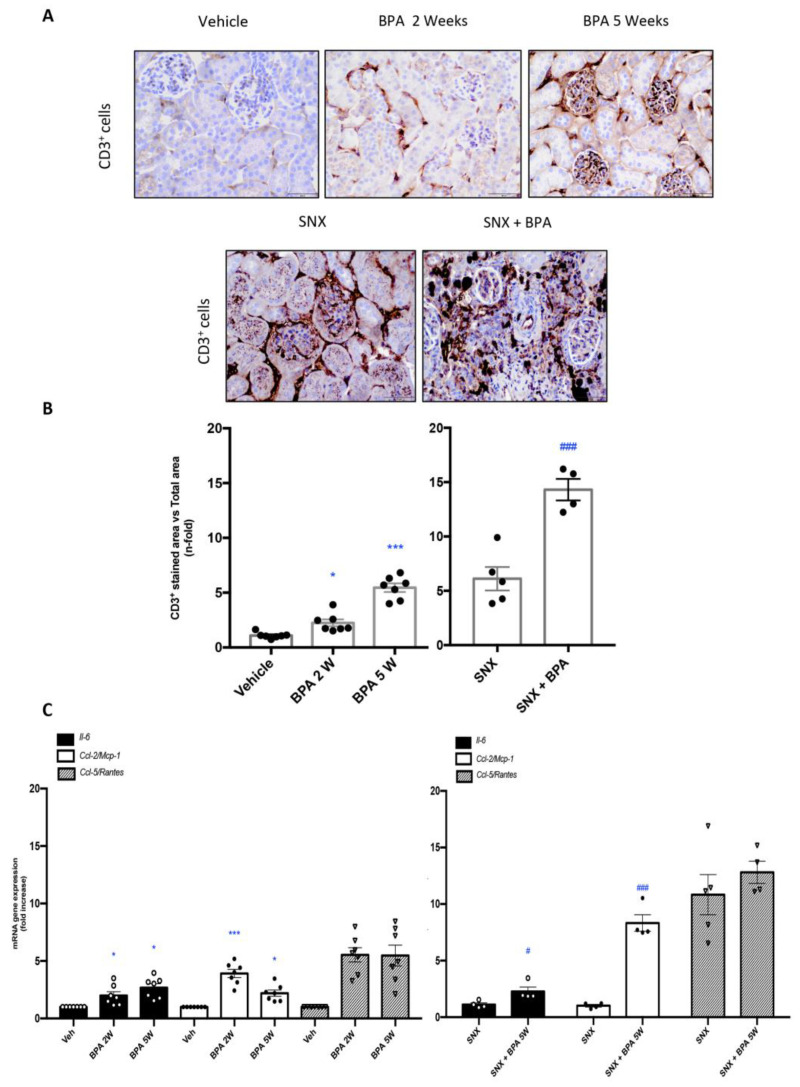
Systemic chronic administration of BPA causes interstitial inflammatory cell infiltration in the kidney and increases renal expression of proinflammatory factors. Animals were intraperitoneal injected with 120 mg/kg/day BPA (5 days a week) or vehicle (corn oil) and sacrificed after 2 or 5 weeks. Some animals were subjected to subtotal nephrectomy (SNX), then treated or not with BPA and sacrificed after 5 weeks. (**A**) Paraffin-embedded kidney sections were stained with an anti-CD3^+^ antibody. Representative immunohistochemistry pictures identifying inflammatory T cell infiltration (CD3+ T lymphocytes). Magnification 200× (scale bar appears in lower right area of the image and represents 50 µm). (**B**) Immunohistochemistry staining quantification expressed as mean of stained area vs. total area ± SEM of 4–7 animals per group. (**C**) Gene expression of *Il-6, Ccl-2 (Mcp-1)* and *Ccl-5 (Rantes)* were evaluated by RT-PCR. Data are expressed as mean ± SEM of 4–7 animals per group. * *p* < 0.05; *** *p* < 0.001 vs. Vehicle, # *p* < 0.05; ### *p* < 0.001 vs. SNX. Comparison between SNX and SNX + BPA groups was performed using a parametric unpaired two-sided *t* test. The analysis of WT vs. BPA-treated groups was done by one-way ANOVA, followed by the Tukey HSD multiple comparison test.

**Figure 3 ijms-22-07189-f003:**
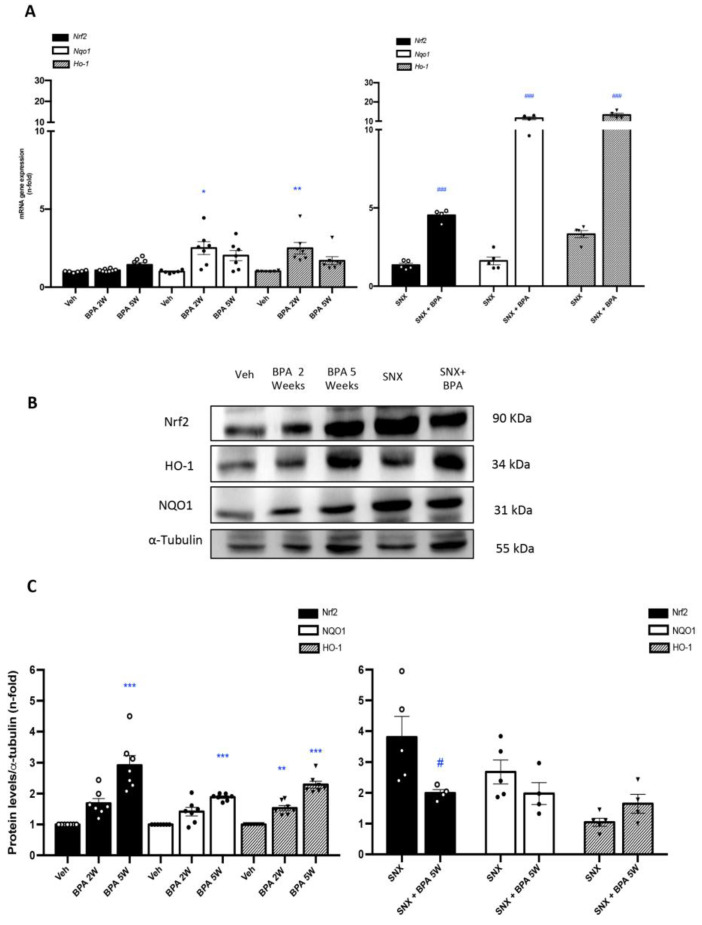
**Systemic chronic administration of BPA activates Nrf2 pathway in SNX model of renal damage.** Animals were intraperitoneal injected with 120 mg/kg/day BPA (5 days a week) or vehicle (corn oil) and sacrificed after 2 or 5 weeks. Some animals were subjected to subtotal nephrectomy (SNX), then treated or not with BPA and sacrificed after 5 weeks. (**A**) mRNA expression of *Nrf2* and Nrf2-regulated genes: *Nqo-1 Ho-1* evaluated by RT-PCR in renal tissue. (**B**,**C**) Representative Western blot image and the quantification of Nrf2, NQO1 and HO-1 levels in renal tissue lysates. All data are presented as the mean ± SEM of 4–7 mice per group, * *p* < 0.05; ** *p* < 0.01; *** *p* < 0.001 vs. Vehicle, # *p* < 0.05; ### *p* < 0.001 vs. SNX. Comparison between SNX and SNX + BPA groups was performed using a parametric unpaired two-sided *t* test. The analysis of WT vs. BPA-treated groups was done by one-way ANOVA, followed by the Tukey HSD multiple comparison Test.

**Figure 4 ijms-22-07189-f004:**
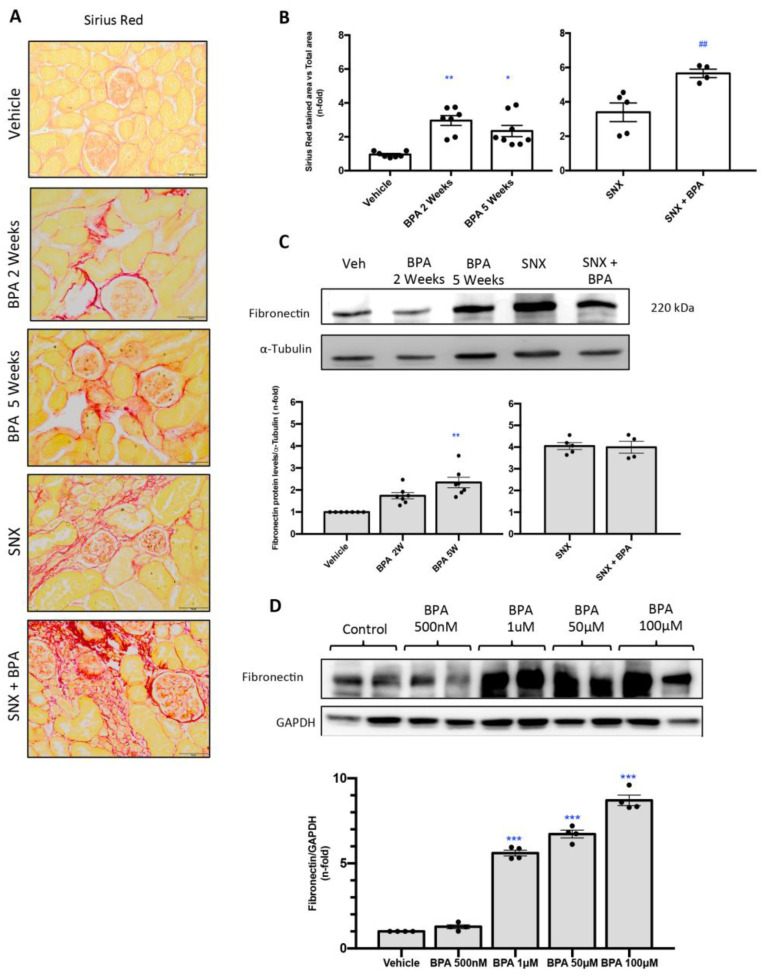
**Systemic chronic administration of BPA induces renal fibrosis in mice**. Animals were intraperitoneal injected with 120 mg/kg/day BPA (5 days a week) or vehicle (corn oil) and sacrificed after 2 or 5 weeks. Some animals were subjected to subtotal nephrectomy (SNX), then treated or not with BPA and sacrificed after 5 weeks. (**A**,**B**) Collagen deposition was evaluated in paraffin-embedded sections by Sirius Red staining; quantification assessed the stained area vs. total area. (**A**) Figures show a representative picture from each group. Magnification 200×. (scale bar appears in lower right area of the image and represents 50 µm). (**B**) The quantification of Sirius Red staining. (**C**) Fibronectin protein levels were evaluated in total renal extracts by Western blot. Figures shows representative mice from each group and the quantification of the Western blot data. Data are expressed as mean ± SEM of 4–7 animals per group. * *p* < 0.05 vs. control; *** *p* < 0.001 vs. control; ## *p* < 0.01 vs. SNX. (**D**) **Treatment with BPA increases ECM proteins production in cultured renal cells.** HK2 cells were treated with BPA at dose of 500 nM, 1, 50 and 100 μM for 48 h. Fibronectin levels were detected by Western blot. Figures shows representative experiment and the quantification of the Western blot. All data are presented as the mean ± SEM of 4 experiments, * *p* < 0.05; ** *p* < 0.01; *** *p* < 0.001 vs. Vehicle, ## *p* < 0.01; vs. BPA. Comparison between SNX vs. SNX + BPA groups was performed using a parametric unpaired two-sided *t* test. The analysis of WT vs. BPA-treated groups was done by one-way ANOVA, followed by the Tukey HSD multiple comparison Test.

**Figure 5 ijms-22-07189-f005:**
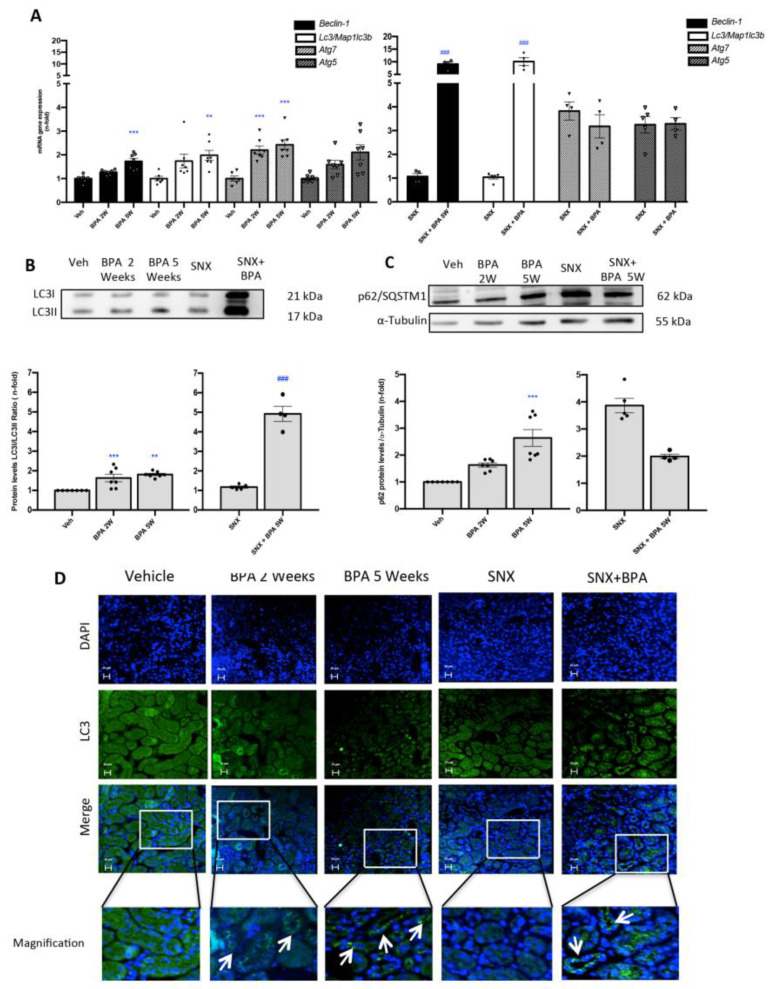
**Systemic chronic administration of BPA blocks autophagosome maturation in experimental SNX model**. Animals were intraperitoneal injected with 120 mg/kg/day BPA (5 days a week) or vehicle (corn oil) and sacrificed after 2 or 5 weeks. Some animals were subjected to subtotal nephrectomy (SNX), then treated or not with BPA and sacrificed after 5 weeks. (**A**) *Beclin-1*, *Atg5, Atg7, Map1lc3b/Lc3b* mRNA levels were assessed by RT-PCR in renal tissue. (**B**,**C**) Representative Western blot and quantification of protein levels of LC3II/I ratio and p62/SQSTM1 in renal tissue lysates. All data are presented as the mean ± SEM of 4–7 mice per group, ** *p* < 0.01; *** *p* < 0.001 vs. Vehicle, ### *p* < 0.001 vs. SNX. (**D**) Immunofluorescence staining localized LC3 expression in renal tubules in all groups with exposure to BPA. Magnification 200×; scale bar represents 20 µm (arrows identify LC3B cytoplasmic staining). Comparison between SNX and SNX + BPA groups was performed using a parametric unpaired two-sided *t* test. The analysis of WT vs. BPA-treated groups was done by one-way ANOVA, followed by the Tukey HSD multiple comparison Test.

**Figure 6 ijms-22-07189-f006:**
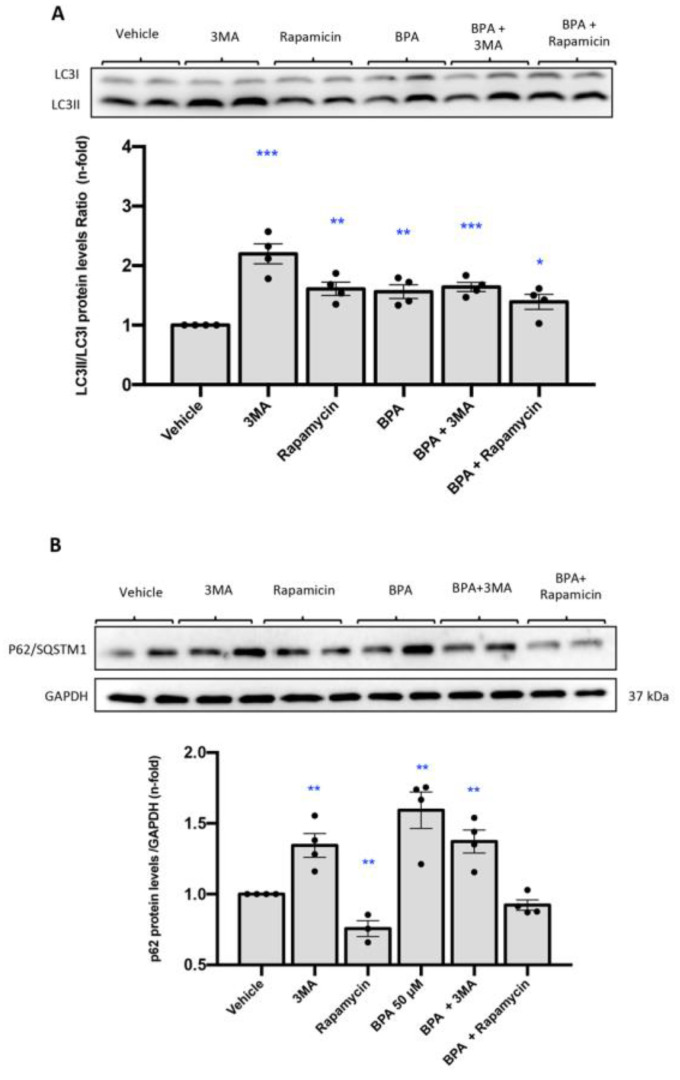
Treatment with BPA causes LC3B accumulation in cultured renal cells compared to 3MA and Rapamycin. (**A**) HK2 cells were treated with BPA at dose of 50 μM for 24 h. The level of LC3II/I was detected by Western blot. LC3II/I protein levels ratio of. BPA treatment induced p62/SQSTM1 accumulation in HK2 cells compared to 3MA and Rapamycin. (**B**) HK2 cells were treated with BPA at dose of 50 μM for 24 h. The level of p62/SQSTM1 was detected by Western blot. p62/SQSTM1 protein levels quantification. All data are presented as the mean ± SEM of 4 experiments, * *p* < 0.05; ** *p* < 0.01; *** *p* < 0.001 vs. control.

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
