# Peer review of "Bisphenol A Modulates Autophagy and Exacerbates Chronic Kidney Damage in Mice"

_ijms, 2021, doi:10.3390/ijms22137189_

Round 1
Reviewer 1 Report
The review of the manuscript 1262261 submitted to International Journal of Molecular Sciences (MDPI)
“Bisphenol A modulates autophagy and exacerbates chronic kidney damage in mice” by Priego A.R. et al.
In their experiment, the authors estimated the harmful influence of the nvironmental factor bisphenol A (BPA) on the structure and function of the kidneys. Studies were conducted both in vitro (using cell cultures of human proximal tubular epithelial cells (HK-2line)) and in vivo (in mice model of 5/6 subtotal nephrectomy; SNX with appropriate sham control).
The authors demonstrated that 5-day administration of BPA to healthy mice, after 2- or 5-weeks induced inflammatory changes in the kidneys along with tubular damage and renal fibrosis with excessive collagen deposition. These changes were exacerbated in animals undergoing SNX. The pathomorphological changes were accompanied by the upregulation of the genes encoding KIM-1 and NGAL-1. Moreover, the administration of BPA induced autophagic flux blockade, which contributes to the progression of kidney damage.
The paper is interesting. However, I have some serious doubts about the methodology and especially the reliability of the statistical analysis of the results:
- The most serious disadvantage of the manuscript is the very poor statistical analysis of the obtained results. It is not known whether the distribution of the quantitative results obtained was normal or the results did not meet the criteria of a normal distribution. How, in turn, were the qualitative data analyzed (e.g. pathomorphological changes?) The authors performed the Mann-Whitney analysis. It is a non-parametric test, equivalent to the Student's t-test for independent samples. This test is used for results that do not have a normal distribution. Therefore, it is not known why it was used in the analysis. In addition, the Mann-Whitney test is based on a "mean ranks" assessment and for this test the median values of the given parameter should be analyzed, not the means. The above reservations cause that the whole statistical inference is uncertain and in principle, unfortunately, disqualifies the manuscript. I would suggest re-analyzing all of the results, starting with examining the distribution of results and selecting the initial, appropriate test and post-hoc one in the second step of the analysis to evaluate the differences in the study groups.
- The layout of the work is atypical, and I would suggest rewriting the text in the following order: Introduction, Materials and methods, Results, Discussion with conclusions, References
- The number of citations seems to be too large; especially in the fragments relating to "multiple" citations.It seems that the standard in original papers is to limit the cited works to the newest, preferably from the last 5 years.The number of all citied articles (141) in the manuscript is rather appropriate for a review.
- The chapter "References" – the list of citied ites is prepared completely contrary to the guidelines of the MDPI Journal and should be fully adapted to the required standards
- In the "Materials and Methods" section, the authors do not mention how they applied the 3R principles that are now standard in experimental work with laboratory animals.
https://www.nc3rs.org.uk/the-3rs
The description of the SNX procedure does not mention both the preparation of the animals for surgery and treatment in the postoperative period.
- How was the number of mice tested in each group determined? Is not the number of 7-8 individuals too small? Did all animals survive the procedure or there were animals lost in the procedure?
Reviewer 2 Report
Major comment
- Need to explain the relevance of the BPA concentration, 120 mg/kg/day, for the kidney disease and human health. The BPA concentration used this experiment is too high, and toxicological level.
Minor comments
- Lines 15, 84, 93, 129, 158, 175, 202, 435: 120mg/kg/day should be 120 mg/kg/day
- Line 84: hematoxilin should be hematoylin
- lines 213, 227, 230: western blot should be Western blot
- line 236: add a space after 100
- line 267: add a space after 200
- line 275: In 2020 Jacobson et al. developed... should be In 2020, Jacobson et al. [54] developed... then delete [54] after KIM1 in line 278
- line 304: delete "a reduction in "
- line 311: ad as after such
- line 337: others should be other?
- line 339: add it before plays
- All references should be followed the journal style, bisphenol a should be bisphenol A, journal names should be adequately abbreviated and italic, titles should be small capitals
- line 715: scientific name of carp should be italic
Round 2
Reviewer 1 Report
Dear Authors and Dear Editor,
Thank you for re-sending the manuscript “Bisphenol A modulates autophagy and exacerbates chronic 2 kidney damage in mice” by Priego A.R.
In its current form, the manuscript has improved significantly. My main complaint in my previous review was related to the unreliable statistical analysis. Currently, the Authors have performed a re-analysis using, in my opinion, properly selected and used statistical methods.
However, my only concern that still exists is the issue of Figures, presenting the results - as the authors' explanations show – in two very different scientific approaches that cannot be analyzed together. The Figures show the results related to the deleterious effect of BPA in healthy mice (with preserved renal function) and the capacity of BPA to worsen the renal damage (evaluating mice with established CKD) - thus, there are two “vehices” groups placed in the Figures.
In my opinion, such presentation of Figures causes that they lose their legibility and the potential reader may have some problems with understanding the results illustrated in this way. Perhaps it would be a better solution to separate the results into separate Figures. On the other hand, however, I also understand the Authors' intentions that it would probably require a significant increase in the total number of Figures and, as a consequence, the total volume of the paper.
Therefore, I leave the issues of the Figures' legibility to be assessed for reconsideration by the Authors and the Editor of the Journal.
